# Upstream open reading frames repress the translation from the *iab-8* RNA

Yohan Frei[1], Clément Immarigeon[1,2], Maxime Revel[1], François Karch[1], Robert K. Maeda[1]*

**1** University of Geneva Department of Genetics and Evolution, Geneva, Switzerland, **2** Molecular, Cellular and Developmental biology department (MCD) Centre de Biologie Intégrative (CBI), Université de Toulouse, Toulouse, France

* robert.maeda@unige.ch

## Abstract

Although originally classified as a non-coding RNA, the *male-specific abdominal (MSA)* RNA from the *Drosophila melanogaster* bithorax complex has recently been shown to code for a micropeptide that plays a vital role in determining how mated females use stored sperm after mating. Interestingly, the MSA transcript is a male-specific version of another transcript produced in both sexes within the posterior central nervous system from an alternative promoter, called the *iab-8* lncRNA. However, while the MSA transcript produces a small peptide, it seems that the *iab-8* transcript does not. Here, we show that the absence of *iab-8* translation is due to a repressive mechanism requiring the two unique 5' exons of the *iab-8* lncRNA. Through cell culture and transgenic analysis, we show that this mechanism relies on the presence of upstream open reading frames present in these two exons that prevent the production of proteins from downstream open reading frames.

## Author summary

The study of genome wide transcriptomes has shown that there are a number of non-coding transcripts that play important biological functions. What keeps these transcripts non-coding is generally thought to be the lack of a suitable open reading frame from which a protein can be translated. However, aside from their non-coding functions, the increased use of techniques like ribosome profiling has shown that many predicted non-coding transcripts are, in fact, bound by ribosomes and also make functional peptides. The *male-specific abdominal* transcript found within the *Drosophila bithorax complex* is one of them. This transcript codes for a small peptide in the male accessory gland that plays a role in sperm usage. However, an alternative version of this transcript, called the *iab-8 lncRNA*, is made in the central nervous system, where it does not seem to produce this peptide. Here, we show that the translation of biologically functional open reading frames can be regulated in different tissues through regulating translation from upstream open reading frames, using the *iab-8* transcript as a model. In doing so, this mechanism could limit potentially detrimental protein misexpression through post-transcriptional means.

**Data Availability Statement:** The authors confirm that all data underlying the findings are fully available without restriction. The data used to generate the figures are within the paper and its Supporting Information files and available on the

open access server, Yareta under a CC 4.0 license. The DOI is: 10.26037/yareta: zkaxzwqtlfbzdljkubhcyzwiai (https://yareta.unige. ch/archives/33c359bd-39ed-454d-8f46- 123a5d50c015).

**Funding:** This work was supported by the Canton of Geneva (R.K.M and F.K.), the Swiss National Fund for Research (http://www.snf.ch/Seiten/ VariationRoot.aspx) grant 31003A_149634 (to F. K.) and grant 310030_192621 (to R.K.M). and the Georges and Antoine Claraz Foundation Foundation (to F.K. and R.K.M.). Salaries for R.K.M. and F.K. were paid by the Canton of Geneva. The salaries of Y.F. and C.I. were paid by the Swiss National Fund for Research (http://www.snf.ch/Seiten/ VariationRoot.aspx) grant 31003A_149634 (to F.K. and R.K.M.). The salary of M.R. was paid by the Swiss National Fund for Research (http://www.snf. ch/Seiten/VariationRoot.aspx) grant 310030_192621 (to R.K.M). The funders had no role in study design, data collection and analysis, decision to publish, or preparation of the manuscript.

**Competing interests:** The authors have declared that no competing interests exist.

## Introduction

Within the *cis*-regulatory region controlling the expression of the two, most-posterior homeotic genes of *Drosophila melanogaster (abd-A* and *Abd-B)*, two transcripts have been identified that differ only at their 5' ends [1–3]. Both transcripts are spliced and polyadenylated but are expressed in different tissues. Originally both of these transcripts were categorized as non-coding. The longer of these two transcripts is called the *iab-8 lncRNA* and is expressed in the posterior central nervous system (CNS) [1,4]. Loss of the *iab-8* ncRNA leads to male and female sterility, largely due to the loss of a miRNA located within its intronic sequences [1] [5,6]. A slightly shorter, second version of this transcript also exists, called *male-specific abdominal (MSA)* [2]. As its name suggests, *MSA* is a male-specific transcript that was found in libraries made from *Drosophila* male abdomen. Later, it was found that this transcript is specifically expressed in a particular cell type of the seminal fluid producing, male accessory gland (AG), called the secondary cells (SCs) [3]. MSA shares all but its first exon with the *iab8* lncRNA (See Fig 1). As such, it also acts as a template to create the same miRNA as the *iab-8* lncRNA (*miR-iab-8*). We have previously shown that the loss of this miRNA in the AG leads to both the abnormal development of the secondary cells and an abnormal post-mating response in females after mating [3]. Although both of these RNAs were originally classified as non-coding and have essential non-coding functionalities, we recently discovered that the *MSA* version of the transcript is actually protein coding [7].

Using ribosome immunoprecipitation and GFP knock-ins, we showed that the *MSA* transcript serves as a template for a small peptide that we call MSAmiP. Through deletion and frameshift mutations, we found that MSAmiP plays a role in regulating sperm usage in fertilized females. Interestingly, although the *iab-8* transcript shares the sORF coding for MSAmiP, using GFP fusions we were never able to detect this peptide in the CNS, where *iab-8* is expressed. Here, we investigate the regulation of protein translation in the *iab-8* transcript. Using GFP reporter constructs in tissue culture cells and transgenic constructs in living flies, we show that the presence of upstream ATG sequences (or upstream open reading frames—uORFs) in the 5' exons of *iab-8* prevent downstream translation from the *iab-8* transcript. Given the conservation of these uORFs in *Drosophila*, it is interesting to note that we can find no overt phenotypic effect on the development or behavior of the fly, when we ectopically express the MSA version of the transcript. We explore the possibility that this repressive mechanism may be a remnant of an ancient method to prevent ectopic activation of the *abd-A* homeotic gene.

## Materials and methods

### S2 cell transfection and fluorescence quantification

S2 cell experiments were performed as in [7]. Basically, ~125 000 S2 cells were placed onto sterilized glass coverslips in 24-well plates, 24 hours prior transfection. The next day, three plasmids (300ng each per well): pActin-Gal4, pUAS-mCherry [7] and an experimental pUAS-plasmid were mixed and transfected with Cellfectin according to manufacturer's directions. All transfections were performed in triplicate. 3 days post transfection, plates were spun and the cells were fixed onto the coverslips with 10% formaldehyde in PBS. The coverslips were then mounted on slides with Vectashield including DAPI (Vector labs, California, USA) and imaged on a Leica SP8 confocal microscope with 10, 20μm step stacks in the green and in the red channel using identical settings (Green channel: HyD3 Laser: 488nm, 0.07%, gain: 100%, Red channel: PMT2 Laser 552nm, 2%, gain: 500). Image quantification was performed in Fiji [8], using a macro described in [7]. Quantification was performed using multiple comparison Kruskal-Wallis test (non-parametric) on the PRISM software (GraphPad, Boston, USA).

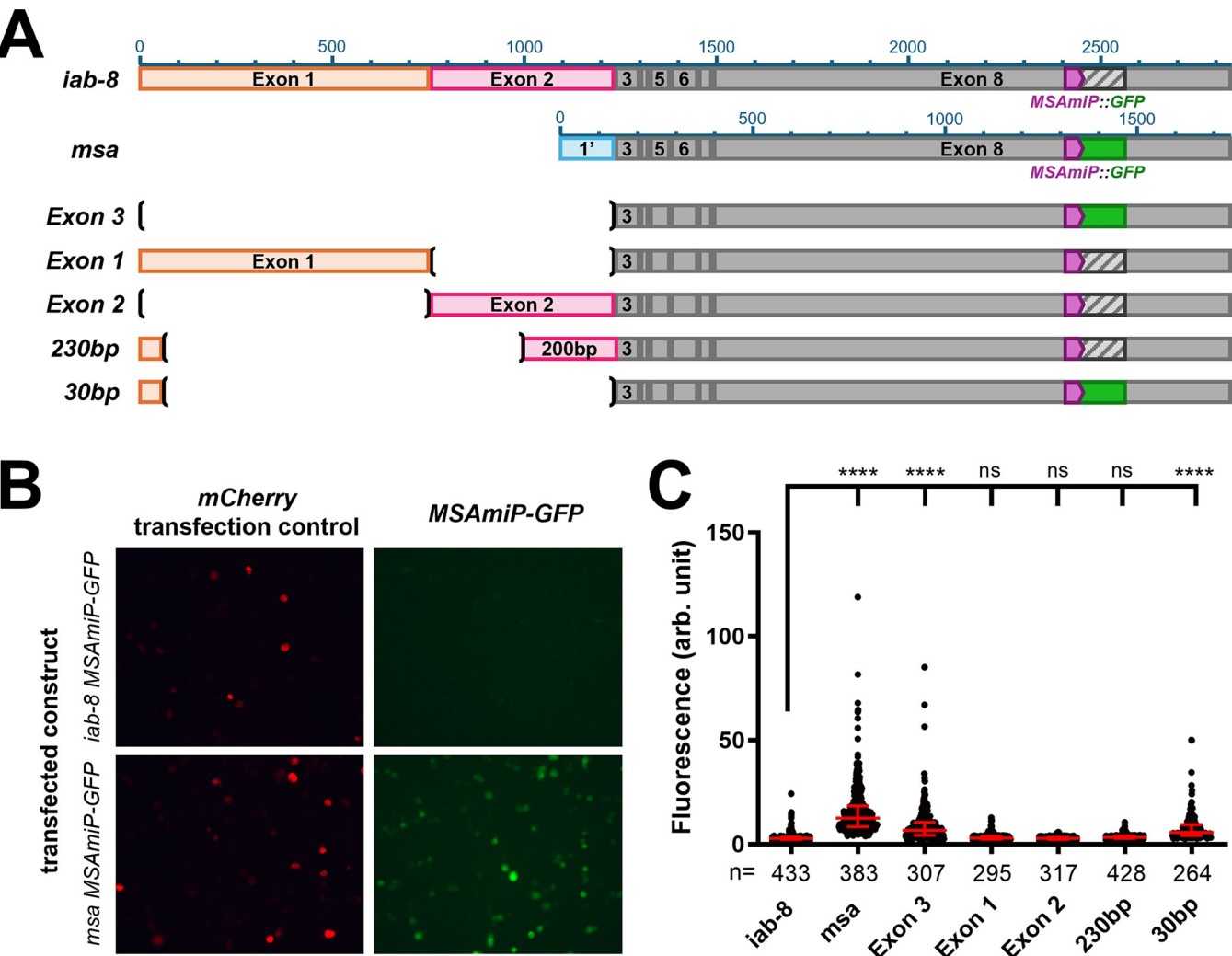

**Fig 1. An element in the 3' area of *iab-8* exon 2 is able to repress downstream translation. A.** *Summary of the results and constructs.* The different constructs tested are schematized with their names indicated on the left. The exons of the *iab-8* transcript are represented by rectangles and labeled accordingly. The coding sequence for MSAmiP is represented in purple with the GFP coding sequence fused to the MSAmiP sequence (without an ATG start codon) indicated by a solid green box for constructs that expressed GFP or by a grey hatched box for construct that did not (see panel C). The sizes in nucleotides are indicated by the scale bars above the first two constructs (*iab-8 and msa* cDNA). The "**(**"and "**)**" represent DNA fragment deletions. **B.** An example of the results from the S2-cell transfections visualized by confocal microscopy using the same settings for each fluorophore. The *iab-8-MSAmiP-GFP* transfection is shown on top and the *msa-MSAmiP-GFP* transfection is shown on the bottom. The left panels show the mCherry control signals, while the right panels show the GFP signals. **C.** Quantification of the fluorescence from the confocal images using the Fiji software and using Prism (bottom left graphic). The X axis lists the different plasmids tested while the y-axis indicates the relative fluorescence of individual cells (arbitrary units). Each dot represents one cell. The errors bars indicate the median with the interquartile range. To determine statistically different levels of expression, we used the multiple comparison Kruskal-Wallis test (non-parametric) in the PRISM software. The **\*\*\*\*** indicate p<0.0001 relative to the *iab-8* construct.

## Plasmid construction, primers and gene fragments

Details on the construction of each of the plasmids used in this study are presented in S1 Text.

## Flies manipulations

The pUASexon1&2, pUASexon1&2ΔATG, pUAS-ORF1-GFP, and pUAS-ORF2-GFP were injected at Bestgene Inc (Chino Hills, CA, USA) into the 68E platform [9].

### CrispR Injections

Guide plasmids and pNProsyattPinPminit were co-injected in equimolar ratio (total concentration: 300ng/μL) in *lig4 vas*::*Cas9;;ry506* flies. $G_o$ flies were crossed with TM2/MKRS balancer stock, and each *ry*+ F1 progeny was recovered and independent stocks were established from each of these flies. Single PCR was performed on each *ry*+ F1 fly using the 5'ryAS-PCR/YR and 3'ryS-PCR/IF primers which amplify the regions flanking the target site only if correctly inserted. Sequencing of the amplified fragment confirmed the insertion.

### Injection in iab8promyf platform

A y1 M{vas-int.Dm}ZH-2A w*;iab8promYF/TM6 stock was first established for our injections. We injected the pexon1-ΔORF1&2, pexon1-ORF1-GFP, pexon1-ORF2-GFP, pexon1-rescue, pexon1-Gal4 plasmids in the iab8promYF line at a concentration of (300ng/μL) in dechorionated embryos. $G_o$ flies were then crossed to TM2/MKRS balancer line and screened for ry- flies, marking an integration event that removes the ry+ marker. The candidate stocks homozygous fertile for the insertion were selected and the regions was PCRed and sequenced to verify fidelity and directionality of the insertions. To screen for single insertion event, we did a PCR with HR5-check/IF primers. For the PCR of the deletion and rescue constructs, we used the HR5-check/ R DON-ex1, while for the Gal4 constructs, we used the HR5-check/Gal4-RT Rev

### Staining of fly tissues

For fixation and staining of accessory glands, we used a protocol provided by Dr. Elodie Prince [10]. Wing disc, embryos and adult tissues were performed as in [11]. All samples were mounted in Vectashield with DAPI. Antibodies against ABD-A: goat polyclonal anti-ABD-A (DH-17) (Santa Cruz biotechnology), monoclonal anti-ABD-A (hybridoma bank), rat anti-ABD-A [12] and Rabbit anti-ABDA [13]. Images were taken on a Zeiss LSM 800 or a Nikon Ni-E confocal microscope.

### Alignment of ORFs present in exons 1 and 2 across different Drosophilids

Genomic regions surrounding exon 1 of the *iab8* ncRNA in non-*melanogaster* species have been identified and downloaded using NCBI's BLAST against the genome assemblies referenced under the BioProject PRJNA675888 [14]. All ORFs longer than 50 amino-acid residues have been extracted from the genomic sequences as well as in the corresponding genomic region in *Drosophila melanogaster*. ORFs presenting high similarities with ORF 1 or 2 from *D. melanogaster* were manually picked, and aligned using the multiple sequence alignment software MUSCLE, available on the sequence analysis application SeaView5 [15] [16]. For several species (*D. biarmipes*, *D. teissieri*, *D. eugracilis*, *D. elegans* and *D. erecta*) two ORFs of interest were found highly resembling the beginning and the end of ORF 2 of *D. melanogaster*. For those species the nucleotide sequences covering both parts were extracted and, when necessary, one or two bases were added after the first STOP codon to reveal both ORFs in one single string of amino-acid that could be aligned. Codon rarity assessment was performed using the %MinMax algorithm at http://www.codons.org/calc.html. *Drosophila* codon usage tables available as a software option.

## Results

Previously, we described the generation of a *Drosophila* line that inserts the *mCherry* coding sequence into exon 3 of the *iab-8* ncRNA within the *Drosophila bithorax complex* [3]. Although RT-PCR performed on embryonic RNA extracts confirmed that the mCherry

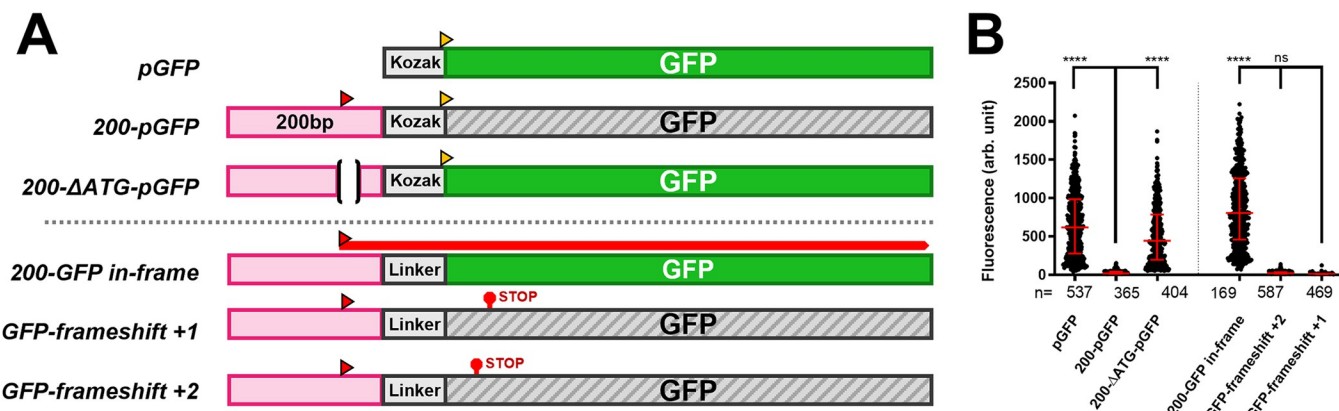

**Fig 2. An ORF in the 200bp fragment represses downstream translation. A.** *Summary of the results and constructs.* The different constructs tested are schematized with their names indicated on the left. The pink rectangles represent the 200 bp fragment being tested. The position of each start codon is indicated by a triangle above the construct. The coding sequence of GFP (without an ATG start codon) is indicated by a solid green rectangle for constructs that express GFP and by a grey hatched box for constructs that did not (see panel B). The 14 bp deletion centered on the ATG sequence in the 200 bp fragment is indicated by the "()" interrupting the pink rectangle. The added Kozak sequences and linkers are represented by labeled boxes. Note that the Kozak sequence here contains an ATG start codon (in yellow), but the linkers lack this element. The three last constructs represent the constructs generated for the translation frame test. They differ in that the 200 bp fragment is reduced by one or two bases to modify the reading frame relative to the GFP sequence. **B.** The graph indicates the relative fluorescence measured (arbitrary units, Y-axis) for each of the construct tested (X-axis). Each dot represents a measured S2 cell. The error bars indicate the median with the interquartile range. Statistical analysis was performed using the multiple comparison Kruskal-Wallis test (non-parametric) on the PRISM software. The **** indicate p<0.0001 relative to the construct indicated with the longer bar.

sequence was properly spliced into the mature *iab-8* RNA, fluorescence was never observed in the CNS where the *iab-8* transcript is normally expressed. However, an *mCherry* signal was detected in the male accessory glands [3]. We now know that this stems from the incorporation of the *mCherry* CDS in an alternative form of the *iab-8* transcript called *male-specific abdominal (msa)* that is expressed in the secondary cells of the male accessory gland.

We wondered why the *mCherry* sequence could be expressed from the *msa* transcript, but not from the *iab-8* transcript and whether this might be of biological importance. We recently showed that the *msa* transcript codes for a micropeptide (MSAmiP) that mediates sperm usage in mated females [7]. Thus, we hypothesized that the lack of mCherry signal could be indicative of a mechanism to prevent MSAmiP from being expressed in the posterior CNS from the *iab-8* transcript.

The main difference between the *msa* and *iab-8* transcripts lies in the first exons. *MSA* starts from a promoter downstream of the *iab-8* promoter and results in the first two exons of *iab-8* being replaced by an alternative first exon. We tested the translation potential of these two transcripts by placing the coding sequence of GFP as an in-frame fusion to the MSAmiP sequence in the context of the full length *MSA* or *iab-8* cDNAs, and transfecting these constructs into S2 cells under the control of a UAS promoter (Fig 1). Co-transfection with a ubiquitous actin-Gal4 plasmid and an mCherry transfection control plasmid showed that GFP expression could be seen from the *MSA* version of the plasmid, but not from the *iab-8* version (Fig 1). Thus, from this initial work, we reasoned that either something in the first two exons of *iab-8* was refractory towards downstream translation or that something in the *msa* first exon promoted downstream translation.

To test if *exon1* and *2* from the *iab-8 lncRNA* contained elements to repress GFP production in S2 cells, we transfected constructs based on the *iab-8* transcript with the *MSAmiP-GFP* fusion but with different portions of exons 1 or 2 deleted (Fig 1). Removal of both exons 1 and 2 from the *iab-8* RNA sequence allows for expression of GFP but the presence of either exon 1 or exon 2 is sufficient to reduce GFP expression in S2 cells. This indicates that redundant

repressive elements capable of preventing *MSAmiP-GFP* expression must be present in the *iab-8* lncRNA. Further dissection of these exons shows that multiple subfragments from exons 1 and 2 are also capable of repressing the expression of the *MSAmiP-GFP* fusion. The smallest fragment found to contain a repressive element in this assay seems to be located in an ~ 200 bp region from exon2. Fusing this ~200 bp fragment directly to GFP, shows that this fragment is capable of reducing GFP expression independent of other elements present in the *iab-8-MSA-miP-GFP* constructs (Fig 2).

Published results have shown that one mechanism that can inhibit translation relies on upstream ORFs inhibiting the translation of downstream ORFs [17, 18] [19] [20]. We wondered if this could be the case with the *iab-8 lncRNA*. Thus, we examined the exon 1 and 2 regions for possible ATG sequences from which upstream translation might emanate. The 200bp fragment from exon 2 contains a single ATG sequence and is preceded by a putative *Drosophila* Kozak sequence [21]. We tested if the removal of this sequence could affect GFP expression in our reporter assay. As seen in Fig 2, removal of the 14 bps around the ATG and Kozak sequence from the 200 bp fragment allows GFP fluorescence levels to reach the level found when transfecting constructs containing the GFP sequence alone.

We next asked if translation is actually initiated from this ATG sequence. To do this, we placed the GFP coding sequence (without its own start codon) in frame with the ATG in the 200 bp fragment (Fig 2B). As there is only one ATG sequence on this fragment and no stop codons in the sequence between the ATG and the GFP CDS, we would expect any GFP fluorescence to stem from translations from this ATG. As negative controls, we also created two frame-shifted constructs that place the GFP in the two alternate reading frames. Transfection of these constructs shows that GFP is only expressed from the construct where GFP is in-frame with the ATG of the 200 bp fragment. Together with the previous results, this shows that the ATG in the 200 bp fragment is required to repress GFP expression when placed upstream of the GFP coding sequence (not in frame), and that this ATG is used to initiate translation.

A dissection of the remaining portions of exons 1 and 2 was similarly performed (Fig 3). For this analysis, the sequence of exons 1 and 2 were first divided into five fragments labeled A-E (with a sixth fragment, F, being the 200 bp fragment described above). Combinations of these fragments were tested for their ability to repress GFP protein production. As seen in Fig 3, repressive activity could be found in Fragments B and D.

Given our results during the examination of the 200 bp fragment (fragment F), we tested the B and D regions for potential translational initiation by fusing an ATG-less GFP coding sequence downstream of the fragments in each of the three reading frames. Figs 3A and 4A–4C show the different uORFs and start and stop codons, color-coded according to a standardized reading frame. To make the analysis easier, fragment B was further divided into two smaller, but overlapping fragments. The $B_{small}$ fragment contains three ATG sequences in two different reading frames but contains no stop codons in these frames. The $B_{big}$ fragment, on the other hand, contains six ATG sequences with no downstream, in-frame stop codons. Thus, using these constructs, we could determine if any start codons were able to initiate translation resulting in GFP expression.

Translation from the $B_{big}$ and D fragments could only be detected when the GFP sequence was placed in frame with the first ATG sequence of each fragment. For the $B_{small}$ fragment, strong levels of GFP expression could also only be detected from constructs where GFP was placed in frame with the first ATG sequence (in red, Fig 4D), Interestingly, however, a low, but significant level of translation could also be found for translation in frame with the second ATG. As the second and third ATGs in $B_{small}$ are the first and second ATGs in in $B_{big}$ that initiate strong translation (Fig 4B and 4C), we hypothesized that, as a rule, translation of upstream

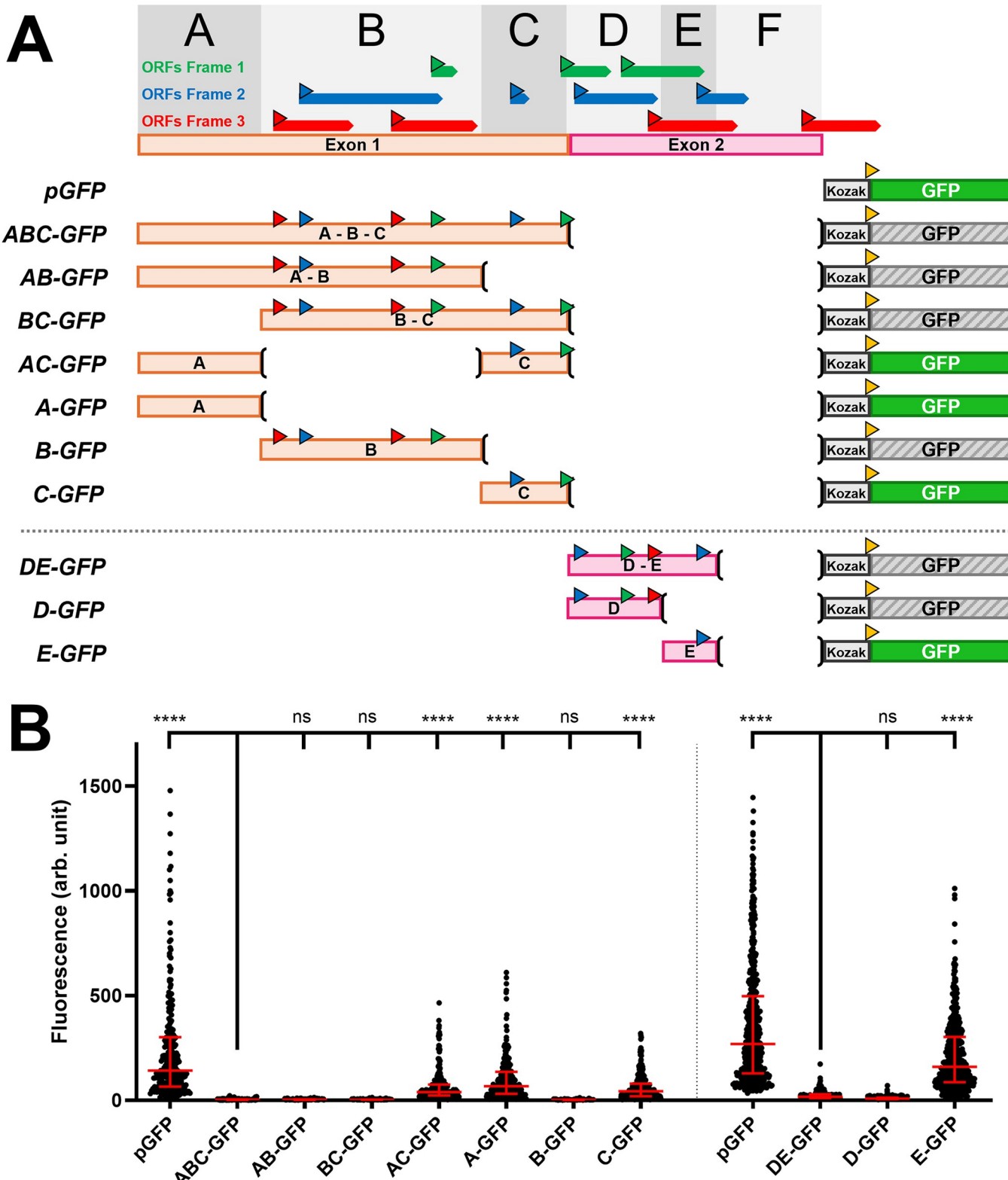

**Fig 3. Multiple uORFs in *iab-8 exon1&2* can repress downstream translation. A.** *Summary of the results and constructs.* The different constructs tested are schematized with their names indicated on the left. Above the constructs are schematic representations of the two firsts exons of the *iab-8 ncRNA* with all of the potential ORFs indicated by the directional boxes, color-coded based on their reading frame relative to the beginning of the transcript. The start codons of each ORF are indicated by colored triangles using the color code established for the ORFs. The *exon-1&2* region was divided into sub-fragments delineated

(Fragments A-F) and labeled in the top panel of **A.** The **(** and **)** represent DNA fragment deletions. The coding sequence of GFP (with the ATG start codon in yellow) is indicated by a solid green rectangle for constructs that express GFP and by a grey hatched box for constructs that did not based on the results displayed in panel B. Kozak sequences are indicated by light grey boxes. **B.** The graph shows the relative fluorescence measured (arbitrary units, Y-axis) for each of the constructs tested (X-axis). Each dot represents a measured S2 cell. The error bars indicate the median with the interquartile range. Statistical analysis was performed using the multiple comparison Kruskal-Wallis test (non-parametric) on the PRISM software. The **\*\*\*\*** indicate p<0.0001 relative to the construct indicated with the longer bar.

ORFs might repress translation initiation from downstream ORFs. Thus, in $B_{small}$, initiation from the first start codon would repress translation from the later start codons, but removal of an upstream start codon, as seen by dividing $B_{small}$ and $B_{big}$, allows for translation initiation to start at the downstream ATG sequence. To test this, we mutated the first ATG codon in $B_{small}$ into a GCC sequence and reexamined the translation of GFP in the three reading frames. Consistent with the repression of downstream ORFs by upstream ORFs, mutation of the first start codon, allowed for an increase in translation initiation from the more-3' start codon (Fig 5B

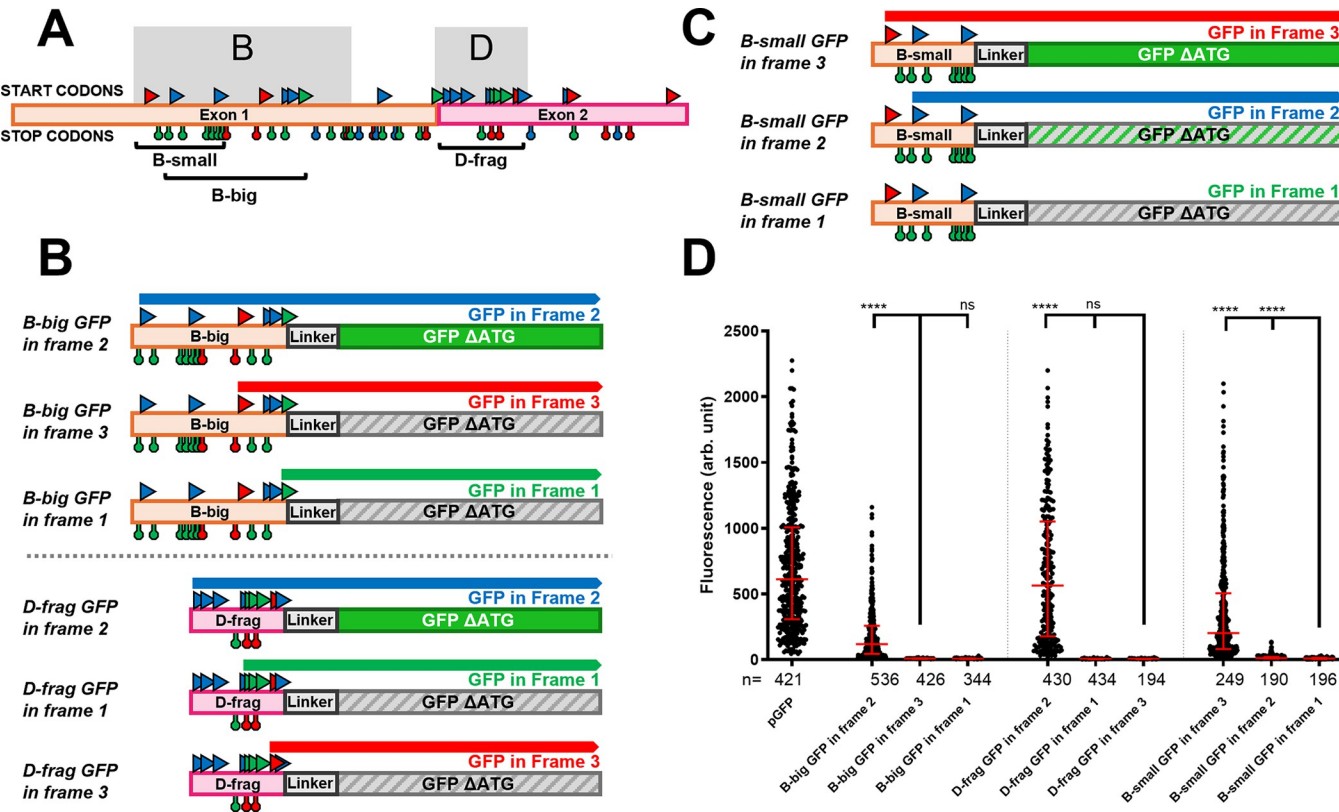

**Fig 4. Context dependent translation from the multiple uORFs in the B and D fragments. A.** A schematic representation of the two firsts exons of the *iab-8* transcript with all of the potential start codons indicated by triangles above the exons and stop codons indicated by hanging octagons below the exons. The start and stop codons are color coded as in Fig 3, based on the reading frames relative to the start of the transcript. The $B_{big}$, $B_{small}$ and D fragments are delineated and labeled. **B. and C.** *Summary of the results and constructs.* The different constructs tested are schematized with their names indicated on the left of each construct. The potential start and stop codons are indicated with a color code consistent with panel A. The coding sequence of GFP without a start codon was placed in each of the three frames to act as a readout for upstream translation initiation. Above each construct, potential open reading frames that are in frame with the GFP sequence are indicated. Constructs where GFP is expressed show the GFP coding sequence as a solid green rectangle, while constructs that do not show GFP expression are shown as grey hatched rectangles. In one case, an intermediate GFP expression was seen and that is shown as a green hatched rectangle (GFP expression levels are based on results shown in **D**). **D.** The graph shows the relative fluorescence measured (arbitrary units, Y-axis) for each of the constructs tested (X-axis). Each dot represents a measured S2 cell. The error bars indicate the median with the interquartile range. Statistical analysis was performed using the multiple comparison Kruskal-Wallis test (non-parametric) on the PRISM software. The **\*\*\*\*** indicate p<0.0001 relative to the construct indicated with the longer bar.

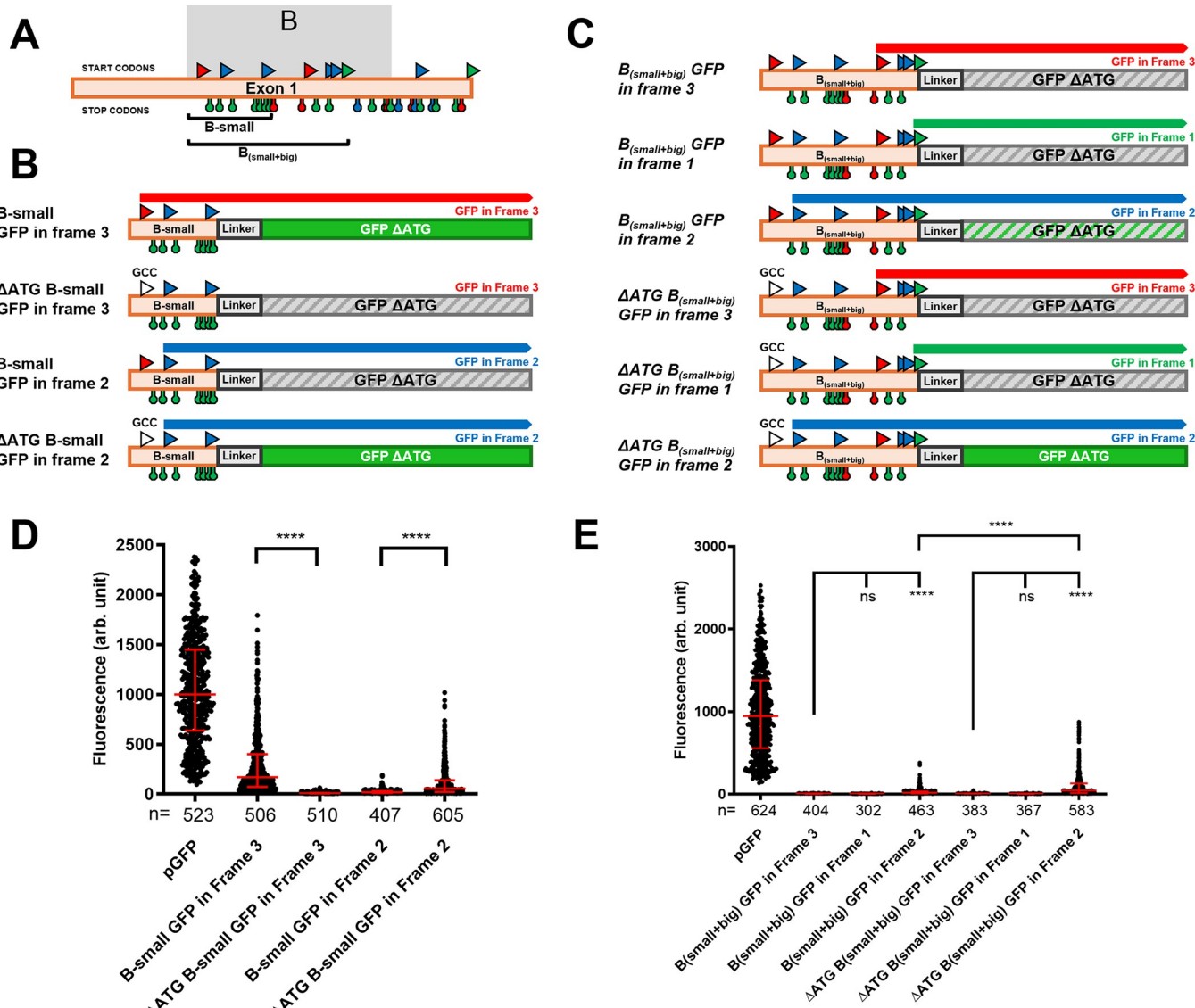

**Fig 5. ORF1 translation represses ORF2 translation. A.** A schematic representation of the first exon of the *iab-8* transcript with all of the potential start codons indicated by triangles above the exons and stop codons indicated by hanging octagons below the exons. The start and stop codons are color coded as in Fig 3 based on the reading frames relative to the start of the transcript. The $B_{small}$ and $B_{small+big}$ fragments are delineated and labeled. **B. and C.** *Summary of the results and constructs.* The different constructs tested are schematized with their names indicated to the left of each construct. The potential start and stop codons are indicated with a color code consistent with panel A. The coding sequence of GFP without a start codon was placed in frame each of the three frames to act as a readout for upstream translation initiation. Above each construct, open reading frame that are in frame with the GFP sequence are indicated. Constructs where GFP is expressed show the GFP coding sequence as a solid green rectangle, while constructs that do not show GFP expression are shown as grey hatched rectangles. Intermediate GFP expression is shown as a green hatched rectangle (GFP expression levels are based on results shown in Panels D and E).). **D. and E.** The graphs show the relative fluorescence measured (arbitrary units, Y-axis) for each of the constructs tested (listed along the X-axis). Each dot represents a measured S2 cell. The error bars indicate the median with the interquartile range. Statistical analysis was performed using the multiple comparison Kruskal-Wallis test (non-parametric) on the PRISM software. The **** indicate p<0.0001 relative to the construct indicated with the longer vertical bar on the graph. To compare the two constructs in frame 2 in **C**, we used the Mann-Whitney test.

and 5D). We also tested this in the context of a longer B fragment that contains both $B_{small}$ and $B_{big}$ ($B_{small+big}$). Because of the presence of two stop codons in-frame with the first start codon in the fragment (identical to the first ATG of $B_{small}$), we would never expect to see translation of GFP from this ATG. However, we did find low levels of translation from constructs with

GFP in frame with the second ATG sequence (the first ATG of B$_{big}$). We hypothesized that the lower level of GFP expression found in this line could be an indication of translation initiating from the first ATG codon inhibiting translation from the second ATG. Thus, we mutated the first ATG codon to GCC and re-examined translation in the three frames. As seen in Fig 5C and 5E, mutating the upstream ATG leads to increased levels of GFP translation in-frame with the second ATG sequence, suggesting again that translation from upstream ATGs inhibits translation from downstream ATGs.

In order to test if the uORF-mediated translational repression mechanism could account for the majority of the repression coming from *iab-8* exons 1 and 2, we mutated all ATG sites in these exons and placed the mutated exons 1 and 2 in front of the GFP coding sequence. Transfecting this construct into cells shows that, in the absence of upstream ATG codons, GFP is able to be translated, even in the presence of the rest of the *iab-8* exons 1 and 2 sequence (Fig 6A). This was also verified in the fly by integrating these constructs into the fly genome. As seen in Fig 6C, no GFP expression can be seen when driving expression of a construct containing exons 1 and 2 in either the secondary cells of the accessory glands or the posterior imaginal discs. However, mutating the ATGs in exons 1 and 2 allow for strong GFP expression in both areas.

The exceptional redundancy of this repression mechanism led us to question why such a mechanism might exist in the *iab-8* transcript. Our first hypothesis was that repression of MSAmiP in the CNS might be important for the viability of the fly. Although we cannot strictly rule out unseen oddities in these flies, thus far, we have not discovered an overt deleterious effect from the ubiquitous expression of the MSAmiP-producing *msa* transcript, in either male or female flies.

An alternative hypothesis is that the upstream open reading frames might actually code for biologically important peptides themselves. Examining the conservation of the coding sequences in 101 *Drosophila* [14] species shows that both of the primary upstream ORFs (corresponding to the first and second start codons above) are conserved in many species (See S1 Fig). Twenty-one of these species show strong conservation of the first 40 codons of the most 5' ORF. Surprisingly, sixteen species show an extensively longer open reading frame, sometimes extending more than 100 codons at the predicted C-terminal end (*D. persimilis* and *D. pseudoobscura*).

The second open reading frame mentioned above is even more conserved, being present in at least 34 species. In *D. melanogaster*, ORF2 is expected to code for an 88 amino acid protein. However, analysis of the conservation between species showed extensive conservation 5' of the predicted ATG. This led us to examine the sequence 5' of the ATG in *D. melanogaster*. This analysis showed that the ORF actually extends far upstream of the predicted ATG of ORF2, beyond the start of the *iab-8* transcript. Four species seem to contain this same 5' extension whereas in seven other species a stop codon seems to have appeared between ORF2 and the new upstream ATG. Overall, this conservation suggests that besides a regulatory role, these translated peptides might have additional biological functions.

In order to start investigating this hypothesis at an experimental level, we used CrispR-mediated homologous recombination to replace exon 1 of the *iab-8* ncRNA with a PhiC31 attP site [22] within the *bithorax complex*. Using this line, we were able to replace the wildtype *iab-8* exon 1 sequence in a second step with an exon 1 sequence lacking the two primary upstream open reading frames (or a control integration using a wild-type exon 1). Although the platform line lacking exon 1 (and the *iab-8* promoter) is homozygous sterile, as predicted for flies lacking the *iab-8* ncRNA (due to a loss of iab-8 miRNA expression), we were unable to find any overt phenotypes in flies where exon 1 was replaced with an exon lacking the ~300 bp containing the two upstream ORFs. Thus, while we cannot rule out more subtle effects or roles

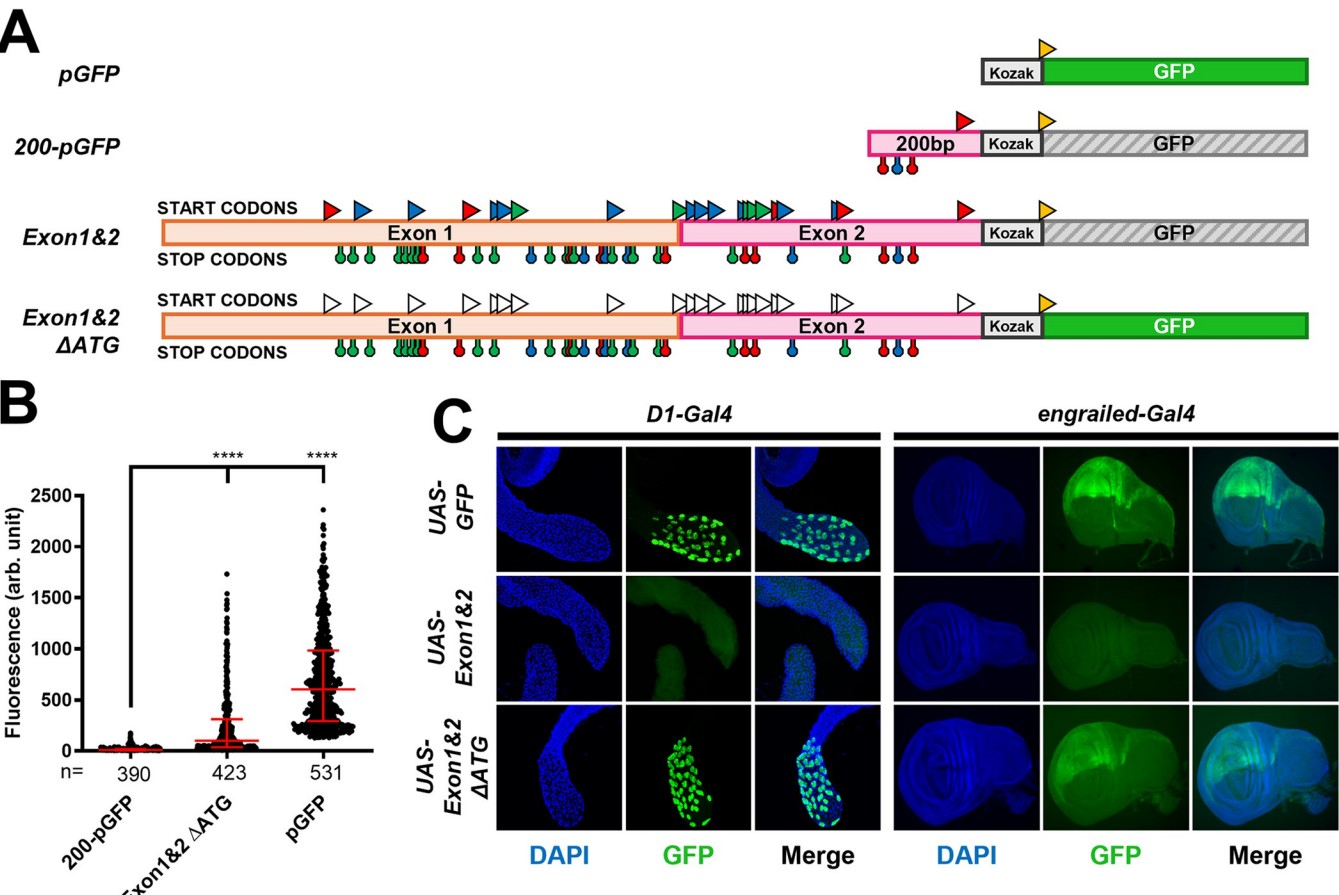

**Fig 6. uORF repression can be monitored in the context of a complete exon 1 and 2 sequence and in flies. A.** *Summary of the results and constructs.* The different constructs tested are schematized with their names indicated to the left of each construct. The potential start (triangles) and stop codons (hanging octagons) are indicated with a color code consistent with Fig 3. Mutated start codons (ATG to GCC) are represented by white triangles. The coding sequence of GFP with a start codon and Kozak sequence was placed downstream of the different forms of exons 1 and 2 of the *iab-8* RNA. Constructs where GFP is expressed show the GFP coding sequence as a solid green rectangle, while constructs that do not show GFP expression are shown as grey hatched rectangles. **B.** The graph shows the relative fluorescence measured (arbitrary units, Y-axis) for each of the constructs tested (listed along the X-axis). Each dot represents a measured S2 cell. The error bars indicate the median with the interquartile range. Statistical analysis was performed using the multiple comparison Kruskal-Wallis test (non-parametric) on the PRISM software. The **\*\*\*\*** indicate p<0.0001 relative to the construct indicated with the longer bar. **C.** GFP immunohistochemistry (IHC) of the constructs (from panel A), in *Drosophila* tissues as visualized through confocal microscopy. GFP expression of the three constructs (listed to the left of the images was driven using the D1-Gal4 driver for the secondary cells of the male accessory gland (left) and the en-Gal4 driver for the posterior wing imaginal discs (right). The first column for each set of images shows DAPI staining to delimit the tissue. The second column shows GFP staining. And the third column shows the merged image. Slight differences in tissue shapes is due to experimental artifacts and could not be attributed to a genotype specific effect.

in other species for these ORFs, our results do not support a role for these peptides in the development or functioning of the fly.

A third alternative reason for developing such a repressive mechanism would be to inhibit the translation of a protein aside from MSAmiP. Previously, we showed that the *iab-8 lncRNA* produces many alternatively spliced products. Some of these transcripts skip the last exon of *iab-8* and splice into a downstream exon of the next downstream gene, *abd-A* [4, 23]. Although these spliced products remove the primary *abd-A* start codon, initiation from a downstream ATG is a possibility. As ectopic expression of *abd-A* in the posterior CNS is known to cause female sterility, this mechanism might be present to ensure that no *abd-A* protein, full length or truncated, is expressed in this area [4] [23].

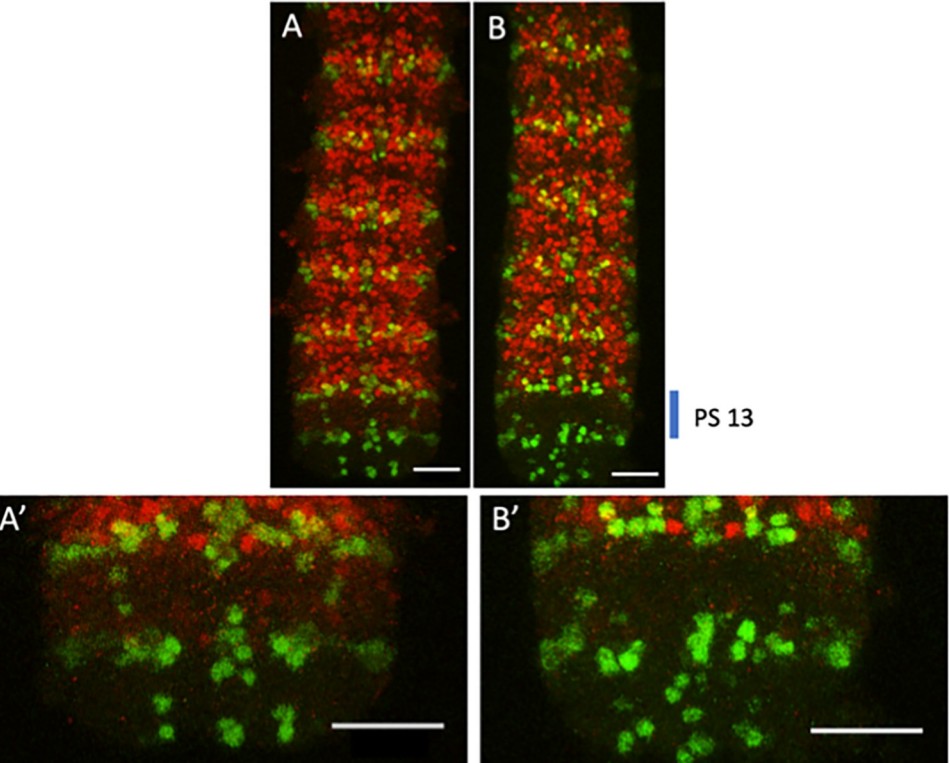

**Fig 7. Loss of ORF 1 and 2 may result in a slight derepression of ABD-A in the CNS of *Drosophila* embryos.** The developing nerve chords were dissected from stage 15 embryos after staining against ABD-A using the goat anti-ABD-A, DH-17 antibody (red) and engrailed as a parasegment marker (green). The location of parasegment 13 is marked on the right of **B. A.** Mutation removing ORF 1 and 2 (ΔORF1&2). **B.** Replacement of the wild-type sequence (wt-rescue). **A'** and **B'** are enlargements of parasegment 13 of **A.** and **B.** respectively. Scale bar = 25μm.

If this third alternative hypothesis is correct, we might expect to see a weak derepression of ABD-A in the posterior CNS. This derepression should be extremely weak due to the multiple redundant elements present in exons 1 and 2 and the remaining presence of mir-iab-8 (which target the *abd-A* transcript). Nevertheless, we investigated the effect of the removal of the ORFs on ABD-A expression by staining the *iab-8* ORF mutant and control embryos for *abd-A* in dissected central nerve chords. Normally, *abd-A* is expressed from parasegment 7 (PS7) to parasegment 12 (PS12) in the early CNS. Removal of the two ORFs from *iab-8* exon 1 seems to result in a very slight ectopic expression of ABD-A in PS13 (Figs 7 and S3). We previously showed that ABD-A protein can be produced in parasegment 13 of the embryonic CNS in embryos lacking the *iab-8* RNA. In those cases, ABD-A expression stems from a derepression of its native promoter and transcript. Here, it seems that the *iab-8* transcript itself is producing what should be a truncated form of ABD-A. This idea is supported by staining with different antibodies to ABD-A, which recognize more N-terminal epitopes (the initial antibody (DH-17) was made to recognize an epitope near the C-terminus (see materials and methods)). Using these other antibodies, ABD-A staining is absent from PS13, but is still visible in PS7-12 (S2 Fig). Unfortunately, we were not able to convincingly quantify this derepression for statistical analysis due to the variability in the staining procedure that resulted in changing background levels (S3 Fig). However, these results suggest that the primary purpose for the upstream ORFs in *Drosophila melanogaster* might be to repress unwanted ABD-A expression that might occasionally result from miss-splicing of the *iab-8* ncRNA.

## Discussion

Molecular biologists have long explored how gene expression levels can be modulated through the control of transcription. And although this has been one of the most studied areas of molecular biology, it represents only one mechanism by which cells can control protein levels. Another method to control the levels of cellular proteins is in the regulation of translation initiation. Since the early studies of Shine-Delgarno and Kozak [24–26], it has long been known that not all start codons promote translation initiation equally. The recent popularity of techniques like ribosome profiling has highlighted an additional level of complexity to this aspect of gene regulation by finding that translation initiation can also be impacted through competition among the sites of translation initiation. Ribosomes are generally loaded at the 5' cap regions of mRNAs. This means that upstream sites of translation initiation will be found by scanning ribosomes before downstream sites. Ribosome profiling studies have shown that there are over 35,000 translated but largely unannotated uORFs present in the ~13500 annotated genes of *Drosophila melanogaster* [27]. Because transcripts with uORFs are markedly less expressed than genes without uORFs, it seems that the presence of uORFs has a negative impact on downstream translation initiation [17]. With the large number of transcripts containing uORFs, the importance of this area of gene regulation seems vastly under-studied by the scientific community.

Here, we show that uORFs present in the first two exons of the *iab-8* transcript reduce translation initiation from downstream ORFs. Past studies where uORFs impact the level of translation of downstream ORFs has made a mechanistic distinction between overlapping and non-overlapping uORFs. In the case of the *iab-8* transcript it seems that we have examples of both types of inhibition. While we were drawn into this work by the effect of translation inhibition by uORFs that seem to be non-overlapping, we found that the numerous "regulatory" uORFs, regulate themselves via overlapping, out-of-frame translation. Indeed, if the site of translation initiation of one uORF is removed, other uORFs begin to be expressed.

While the interactions between overlapping ORFs are believed to occur via physical interference of translating ribosomes and with downstream translation initiation, the control of translation initiation from non-overlapping, downstream ORFs seems to be mechanistically more complicated. Studies have shown that ribosome reinitiation downstream of a translated sequence is somewhat inefficient, but does occur in many places. For example, it has been shown that the yeast GCN4 transcript contains 4 uORFs 5' to the GCN4 coding sequence that regulate the level of GCN4 translation [19]. On the GCN4 transcript, translation of the first uORF is generally initiated. Upon termination, it seems that the 40S ribosomal subunit stays bound to the transcript and continues to scan for a downstream initiation site. The frequency of downstream reinitiation is dependent on the availability of factors that make up the ribosome ternary complex and is impacted by cellular stress. Under stress conditions, these factors are less available and causing the 40S subunit to bypass the additional inhibitory uORFs to eventually find the GCN4 coding sequence, by which time it has formed the ternary complex to initiate translation. Under non-stress conditions, the four uORFs are translated and upon termination, the 40S subunit falls off of the transcript.

It is still somewhat vague as to what dictates the decision of a 40S ribosomal subunit to continue scanning a transcript after translation termination. One important aspect seems to be its ability to progress. For example, the AdoMetDC1 gene of Arabidopsis thaliana that codes for S-adenosylmethionine decarboxylase, an enzyme important in the biosynthesis of spermidine [28]. The AdoMetDC1 transcript contains two, overlapping uORFs. After translation of the major uORF, the ribosome stalls at the stop codon and eventually releases the transcript. In this case, the small uORF codes for a peptide that binds to spermidine to cause the ribosome

stalling. When spermidine is low, the stalling does not occur and the ribosome remains bound to the transcript to reinitiate at the downstream AdoMetDC1 initiation site [28]. A similar type of regulation seems to happen with the CPA1 transcript in yeast that is stalled by high levels of arginine, sensed by the uORF nascent peptide [29] [30] [31]. These examples of stalling ribosomes led us to think about potential mechanisms by which the *iab-8* uORFs might repress downstream non-overlapping ORFs. Although we have not tested if ribosomes stall on the iab-8 transcript, we decided to examine the codon usage of the uORFs with the idea that a preponderance of rare codons might also slow ribosome progression. This analysis shows that the first ORF does indeed contain a large number of rare codons. Using the %MinMax online codon usage tool [33] indicates that 100% of the codons used in this uORF (and 65% of the codons used in the second uORF) are non-optimal. Thus, it may be the concentration of rare codons that might be the mechanism by which this uORF prevents downstream translation.

Based on our results, there seem to be conserved ORFs in the *iab-8* RNA that prevent downstream translation. This conservation suggests that either these ORFs code for important peptides or that the prevention of downstream translation is important for the functioning of the organism. Given that the related MSA transcript makes a biologically important peptide from it shared 3' exon, it seems likely that the translational repression of this peptide may be the biologically important role of these uORFs. Although this may be true at an evolutionary scale, we have not been able to discover an overt phenotype associated with ectopic expression of the MSA transcript. Thus, production of the MSAmiP outside of the male accessory gland does not seem to be a problem.

Previously, our genetic dissection of the *iab-8* ncRNA has indicated that the primary function of the *iab-8* transcript is the production of the *iab-8* miRNA [4,23]. This miRNA is important for repressing the translation of many target genes, including *abd-A*, *Ubx* and their essential cofactors *homothorax* and *extradenticle* [1] [5] [6] [32] [4]. The position of its promoter within the *iab-8* region of the *Drosophila bithorax complex* allows it to be expressed in the posterior of the embryo where it can restrict posterior expression of more-anterior hox genes. We have previously noted that its placement just upstream of one of its primary target genes may be indicative of an ancestral repressive mechanism to silence *abd-A*, requiring its presence upstream of its target gene [23]. This is consistent with both our previously described finding of transcriptional interference on the *abd-A* promoter by the *iab-8* transcript [4,23], and the concept of posterior dominance within the hox complexes (where posterior/upstream genes function over anterior/downstream genes). But this type of regulation can lead to a potential problem when a ncRNA is made to be the agent of transcriptional interference, as alternative splicing can actually lead to unwanted protein synthesis. In the case of *iab-8*, we have previously shown that the mechanism of transcriptional interference actually requires that the *iab-8* transcript extends into the *abd-A* transcription unit and that these transcripts splice into *abd-A* exons [23]. As these transcripts then could create truncated but potentially active ABD-A protein, the repressive transcriptional interference mechanism might actually prove futile. Interestingly, this problem is avoided by simply having upstream open reading frames in the transcript to inhibit potential downstream translation initiation. In other words, using coding genes to inhibit transcription of downstream genes would avoid this problem.

While we believe that the uORFs in the *iab-8* transcript might be used to prevent the creation of spurious ABD-A protein, this does preclude the coded peptides from having a biological function in their own right. Indeed, the conservation of the uORFs support this idea. Thus far, we have not been able to discover an overt phenotype associated with the loss or misexpression of these peptides. However, this is not unexpected and likely reflects the poor resolution of our lens when examining phenotype; subtle behavioral phenotypes or slight changes in neural connectivity, for example, would go largely unnoticed by our analysis. Furthermore, we

cannot rule out that these peptides may play prominent roles in other closely related *Drosophila* species that have been mostly lost in *melanogaster*. Thus, discovering a function for these peptides will require many additional experiments. With the growing number of uORFs being discovered, a more comprehensive method to quickly analyze phenotypes may be required to unlock the vast number of potential biological functions of uORFs.

## Supporting information

**S1 Text. Supplementary Methods.**
(DOCX)

**S1 Fig. Open Reading Frames conservation in the iab-8 ncRNA.** Amino acid alignment results for ORF 1 (**A.**)**,** and ORF 2 (**B.**) across 44 Drosophilid species. The 'X' characters in the amino acids sequences indicates that one or two additional bases have been added to the nucleotide sequence to reveal the conservation of both the beginning and the end of ORF 2 on two different reading frames in some species (*teissieri*, *biarmipes eugracilis*). Italicized characters indicate residues that are not part of any open reading frame, located after a STOP codon and before a following Methionine. C. D. are closer looks at the alignments of the main parts of ORF1 (**C**) and ORF2 (**D**). The alignments were obtained using the muscle algorithm available on the SeaView sequence analysis application. Species names and phylogeny [37] are indicated at the left of each alignment. Amino acid residues are color coded according to the legend at the bottom of the figure.
(TIFF)

**S2 Fig. Staining against ABD-A using N-terminal antibodies in the CNS of Drosophila embryos.** The developing nerve chords were dissected from stage 15 embryos after staining against ABD-A using either a rat anti-ABD-A antibody or a rabbit anti-ABD-A antibody. The location of parasegment 11–13 are marked on the left.
(TIFF)

**S3 Fig. Staining against ABD-A using a C-terminal antibody in the CNS of Drosophila embryos.** The developing nerve chords were dissected from stage 15 embryos after staining against ABD-A using the goat anti-ABD-A, DH-17 antibody (red) and engrailed as a parasegment marker ([40]). The top six images are of posterior nerve chords dissected from an iab-8 rescue line and the bottom six are examples dissected from an iab-8 Exon 1+2 Delta ATG line.
(TIFF)

**S1 Data. This file contains the data with which the graphs in Fig 1 were generated.** The data is presented in excel format. The numbers in the excel files are quantifications of the fluorescence of cells transfected with the construct indicated on the top line of each column. The second sheet and subsequent sheets contain the results of statistical tests run on the data using the Prism Statistical Analysis software package.
(XLSX)

**S2 Data. This file contains the data with which the graphs in Fig 2 were generated.** The data is presented in excel format. The numbers in the excel files are quantifications of the fluorescence of cells transfected with the construct indicated on the top line of each column. The second sheet and subsequent sheets contain the results of statistical tests run on the data using the Prism Statistical Analysis software package.
(XLSX)

**S3 Data. This file contains the data with which the graphs in Fig 3 were generated.** The data is presented in excel format. The numbers in the excel files are quantifications of the fluorescence of cells transfected with the construct indicated on the top line of each column. The second sheet and subsequent sheets contain the results of statistical tests run on the data using the Prism Statistical Analysis software package.
(XLSX)

**S4 Data. This file contains the data with which the graphs in Fig 4 were generated.** The data is presented in excel format. The numbers in the excel files are quantifications of the fluorescence of cells transfected with the construct indicated on the top line of each column. The second sheet and subsequent sheets contain the results of statistical tests run on the data using the Prism Statistical Analysis software package.
(XLSX)

**S5 Data. This file contains the data with which the graphs in Fig 5 were generated.** The data is presented in excel format, named according to the figure it is used in. The numbers in the excel files are quantifications of the fluorescence of cells transfected with the construct indicated on the top line of each column. The second sheet and subsequent sheets contain the results of statistical tests run on the data using the Prism Statistical Analysis software package.
(XLSX)

**S6 Data. This file contains the data with which the graphs in Fig 6 were generated.** The data is presented in excel format. The numbers in the excel files are quantifications of the fluorescence of cells transfected with the construct indicated on the top line of each column. The second sheet and subsequent sheets contain the results of statistical tests run on the data using the Prism Statistical Analysis software package.
(XLSX)

## Author Contributions

**Conceptualization:** Yohan Frei, François Karch, Robert K. Maeda.

**Data curation:** Yohan Frei.

**Formal analysis:** Yohan Frei, Clément Immarigeon, Maxime Revel, François Karch.

**Funding acquisition:** François Karch, Robert K. Maeda.

**Investigation:** Yohan Frei, Clément Immarigeon, Maxime Revel, François Karch, Robert K. Maeda.

**Methodology:** Yohan Frei.

**Project administration:** Yohan Frei, Robert K. Maeda.

**Resources:** Clément Immarigeon.

**Supervision:** François Karch, Robert K. Maeda.

**Validation:** Yohan Frei.

**Visualization:** Yohan Frei, Maxime Revel, François Karch.

**Writing – original draft:** Yohan Frei, Robert K. Maeda.

**Writing – review & editing:** Clément Immarigeon, François Karch, Robert K. Maeda.

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
