## [Decision Letter · Decision Letter 0]

24 May 2024

Dear Dr Maeda,

Thank you very much for submitting your Research Article entitled 'Upstream open reading frames repress the translation of a micropeptide involved in sperm competition' to PLOS Genetics.

The manuscript was fully evaluated at the editorial level and by independent peer reviewers. The reviewers appreciated the attention to an important problem. All three reviewers praised the thorough analysis of *iab-8* uORF repressive effects on downstream ORFs using GFP reporters in S2 cells. However, there were substantial concerns. For example, Reviewer 2 indicates the lack of evidence that MSAmiP is derepressed in the brain after mutating upstream uORFs, and whether brain-expressed MSAmiP has deleterious effects. In addition, all three reviewers indicated issues with the readability of figures and text in the manuscript. Based on the reviews, we will not be able to accept this version of the manuscript, but we would be willing to review a much-revised version. We cannot, of course, promise publication at that time.

If you decide to revise the manuscript for further consideration at PLOS Genetics, please aim to resubmit within the next 60 days, unless it will take extra time to address the concerns of the reviewers, in which case we would appreciate an expected resubmission date by email to plosgenetics@plos.org.

We are sorry that we cannot be more positive about your manuscript at this stage. Please do not hesitate to contact us if you have any concerns or questions.

Yours sincerely,

Justin Bosch

Guest Editor

PLOS Genetics

Gregory P. Copenhaver

Section Editor

PLOS Genetics

Reviewer's Responses to Questions

**Comments to the Authors:**

Reviewer #1: The study investigates the translational regulation of a small peptide, MSAmiP, in Drosophila melanogaster. The study reveals that the male-specific abdominal (MSA) RNA, initially classified as non-coding, actually encodes for MSAmiP, which is produced in the male accessory gland and is crucial for sperm competition. The researchers discovered that the lack of MSAmiP synthesis in the central nervous system (CNS) is due to upstream open reading frames (uORFs) present in the iab-8 lncRNA, which is transcribed from an alternate promoter. Through cell culture and transgenic analysis, the authors demonstrate that these uORFs prevent the production of protein from downstream ORFs. The findings suggest that uORFs play a role in regulating gene expression and may have evolved to prevent unwanted translation of certain transcripts, such as those resulting from alternative splicing. The study provides insights into post-transcriptional regulation's complexity and non-coding RNAs' role in gene expression control. Overall, the experiments are solid and support their conclusion. My concerns are listed below.

1. the illustration of different reporter constructs in Figures 2-5 is hard to understand, especially the different reading frames of start codons and GFP reporters.

2. While the authors have provided a detailed dissection of uORF-mediated translation regulation for MSAmiP, similar mechanisms of regulating protein translation by switching isoforms with different numbers of uORFs have been proposed by several groups as reviewed here (DOI: 10.1016/j.tibs.2019.03.002). It is better to discuss connections with previous studies.

3. "But, this begs the question: were the upstream open reading frames added to the iab-8 RNA to prevent spurious abd-A expression or are they perhaps remnants of a coding gene now potentially in the process of degeneration? The limited conservation of the ORFs may support the latter." Suppose the uORFs in exons 1 and 2 are regulatory. In that case, there is no need for strong conservation for ORF sequence, and even the position of uORF start codons does not matter as long as there are uORFs with strong translatability. Therefore, I do not see why the latter scenario is more likely.

4. To further demonstrate that this is a conserved regulatory mechanism, could the authors show that the iab-8 and MSA transcript isoforms are conserved in other Drosophila species with public data?

5. "we showed that the MSA transcript serves as a template for a small peptide of 11-20 amino acids that we call MSAmiP". What is the exact length of MSAmiP? Is it 11 or 20 amino acids, or any value between 11~20?

Reviewer #2: In their manuscript "Upstream open reading frames repress the translation of a micropeptide involved in sperm competition" Maeda and colleagues explored the genetic regulation of the msa micropeptide (MSAmiP) from Drosophila melanogaster. This micropeptide is not produced in the central nervous system even though the iab-8 lncRNA transcript contains the MSAmiP sequence and is expressed in this tissue. The authors successfully dissected important sequences in exons 1 and 2 of the iab-8 lncRNA transcript that allow for the repression of the GFP fluorescence of a GFP tagged MSAmiP reporter gene and show that upstream ORFs are required to disrupt the fluorescence of GFP in different reporter genes by using S2 cells. Finally, they showed that exons 1 and 2 are sufficient to disrupt the fluorescence of a GFP reporter in vivo, and that mutagenizing the start codons in these exons allows GFP to be properly expressed.

Overall, the authors provide good evidence for their claims with a few caveats mentioned below. However, the main concern is that the in vivo relevance of the regulation is limited, and concerned to UAS constructs. Because they have a genetic engineering platform, I wonder if it is feasible to show that these upstream exons, or indeed specific ATG, are truly responsible to prevent MSA peptide expression in CNS. And, is MSA peptide expression in CNS deleterious? (E.g., is this regulatory mechanism necessary?)

- When analyzing ORFs upstream of exon 2 by themselves (sections A,B,C,D and E in Figure 3) it should be noted that even if they can’t repress GFP in this assay, it is possible that they could still be repressive of the MSAmiP in vivo, since it could be that there are ORFs transcribing with GFP in frame and forming a longer protein that can still fluoresce but in vivo there might be a stop codon downstream that also inhibits the translation of the MSAmiP. Taking this into consideration and adding this information to the iab-8 lncRNA models is crucial to differentiate between translation inhibition due to an upstream uORF and a disruption of the MSAmiP due to the formation of a frameshifted MSAmiP.

- Referring to Figure 7 the authors mention they were “not able to convincingly quantify this derepression for statistical analysis due to the variability in the staining procedure that resulted in changing background levels” if this is the case, they should include at least a few images of each condition to show the variability so that the reader can make their own conclusions.

- This may seem like a minor comment, but I believe it is a substantial issue. The figures are very difficult to read and understand. Fig1-5 are kind of confusing layout, it is a challenge to understand how the different reporter constructs and regions in different figures relate to each other, and what the construct names reflect. The immunostainings do not have even the markers labeled (Fig7). As a reader, this made it very difficult to understand their findings and conclusions.

Other minor comments about the manuscript:

- The 30 bp fragment of exon 1 is not mentioned in the text but has prominent activity change in Figure 1.

- Authors say about the results in Figure 5C “Based on this, we hypothesized that, as a rule, upstream ORFs (uORFs) might repress downstream ORFs.”. This is one general proposal for uORF function, but they are writing it in a way that it is not quite meaningful. I think they are suggesting a competitive impact of translational initiation at a downstream main ORF, if the uORF is translated. But there could be other models at play, and this is not a biochemical mechanistic study. So I suggest to rephrase this.

- When showing fluorescence the Y axis should be more descriptive of what this fluorescence is (per cell, per slide, per pixel, etc). At least it should say “Fluorescence” instead of “Fluo”.

- The order of Figure 5B and 5C should be swapped for readability

Reviewer #3: Review is an attachment

**Have all data underlying the figures and results presented in the manuscript been provided?**

Reviewer #1: Yes

Reviewer #2: Yes

Reviewer #3: Yes

PLOS authors have the option to publish the peer review history of their article (what does this mean?). If published, this will include your full peer review and any attached files.

Reviewer #1: **Yes: **Jian Lu

Reviewer #2: No

Reviewer #3: No

---

## [Decision Letter · Decision Letter 1]

28 Aug 2024

Dear Dr Maeda,

We are pleased to inform you that your manuscript entitled "Upstream open reading frames repress the translation from the iab-8 RNA" has been editorially accepted for publication in PLOS Genetics. Congratulations!

Yours sincerely,

Justin Bosch

Guest Editor

PLOS Genetics

Gregory P. Copenhaver

Section Editor

PLOS Genetics

Comments from the reviewers (if applicable):

Reviewer's Responses to Questions

**Comments to the Authors:**

Reviewer #1: The authors have done a good job revising the manuscript, and all my concerns have been addressed.

Reviewer #3: The authors have addressed all my concerns, particularly as it pertains to editing suggestions. The updated figures have made data interpretation much easier and the description of the data and discussion have also been made clearer. Based on these, I recommend the manuscript for publication.

**Have all data underlying the figures and results presented in the manuscript been provided?**

Reviewer #1: Yes

Reviewer #3: Yes

PLOS authors have the option to publish the peer review history of their article (what does this mean?). If published, this will include your full peer review and any attached files.

Reviewer #1: No

Reviewer #3: No

**Data Deposition**

http://datadryad.org/submit?journalID=pgenetics&manu=PGENETICS-D-24-00268R1

**Press Queries**

---

## [Editor Report · Acceptance letter]

17 Sep 2024

PGENETICS-D-24-00268R1 

Upstream open reading frames repress the translation from the iab-8 RNA 

Dear Dr Maeda, 

We are pleased to inform you that your manuscript entitled "Upstream open reading frames repress the translation from the iab-8 RNA" has been formally accepted for publication in PLOS Genetics! Your manuscript is now with our production department and you will be notified of the publication date in due course.

With kind regards,

Lilla Horvath

PLOS Genetics

On behalf of:
